# Protein Kinase A Distribution in Meningioma

**DOI:** 10.3390/cancers11111686

**Published:** 2019-10-29

**Authors:** Antonio Caretta, Luca Denaro, Domenico D’Avella, Carla Mucignat-Caretta

**Affiliations:** 1Department of Food and Drug, University of Parma, 43100 Parma, Italy; antonio.caretta@unipr.it; 2National Institute of Biostructures and Biosystems, 00136 Roma, Italy; 3Department of Neuroscience, University of Padova, 35121 Padova, Italy; luca.denaro@unipd.it (L.D.); domenico.davella@unipd.it (D.D.); 4Department of Molecular Medicine, University of Padova, 35131 Padova, Italy

**Keywords:** meningioma, Protein kinase A, cAMP

## Abstract

Deregulation of intracellular signal transduction pathways is a hallmark of cancer cells, clearly differentiating them from healthy cells. Differential intracellular distribution of the cAMP-dependent protein kinases (PKA) was previously detected in cell cultures and in vivo in glioblastoma and medulloblastoma. Our goal is to extend this observation to meningioma, to explore possible differences among tumors of different origins and prospective outcomes. The distribution of regulatory and catalytic subunits of PKA has been examined in tissue specimens obtained during surgery from meningioma patients. PKA RI subunit appeared more evenly distributed throughout the cytoplasm, but it was clearly detectable only in some tumors. RII was present in discrete spots, presumably at high local concentration; these aggregates could also be visualized under equilibrium binding conditions with fluorescent 8-substituted cAMP analogues, at variance with normal brain tissue and other brain tumors. The PKA catalytic subunit showed exactly overlapping pattern to RII and in fixed sections could be visualized by fluorescent cAMP analogues. Gene expression analysis showed that the PKA catalytic subunit revealed a significant correlation pattern with genes involved in meningioma. Hence, meningioma patients show a distinctive distribution pattern of PKA regulatory and catalytic subunits, different from glioblastoma, medulloblastoma, and healthy brain tissue. These observations raise the possibility of exploiting the PKA intracellular pathway as a diagnostic tool and possible therapeutic interventions.

## 1. Introduction

Meningiomas comprise the largest number of central nervous system (CNS) tumors, accounting for more than one-third of all CNS tumors, with a female:male ratio of about 2:1 [1]. Despite their number, meningiomas are much less studied compared to other tumors, presumably because of their less aggressive phenotype, unlike the most aggressive glioblastoma (GBM). Presumably, meningioma originates from arachnoidal cap cells located along venous sinuses. The recent 2016 World Health Organization classification of CNS tumors did not revise the classification of 16 existing meningioma entities, except for the inclusion of invasiveness as a diagnostic criterion for Grade II atypical meningioma [2], while for other CNS tumors, molecular characterization has been included in the diagnostic criteria. Deregulation of signal transduction mechanisms can be detected in almost all tumor cells, and aberrant signal transduction mechanisms involved in cytoskeleton regulation are presumably present in many meningiomas. In this sense, meningiomas may prove to be of interest because of the involvement of Merlin, a protein encoded by the NF2 gene, which is mutated or lost in roughly one half of meningioma patients [3]. Merlin, also known as schwannomin, is apparently devoid of any catalytic or DNA binding activity, but is presumably involved in cytoskeleton dynamics, by binding actin, and in cell shape control, contributing to contact-dependent inhibition of proliferation [4]. As expected, this tumor suppressor protein interacts extensively with many signal transduction mechanisms, including the cyclic adenosine-monophosphate (cAMP) pathway; noticeably, it can also function as an A-kinase anchoring protein [5,6].

The cAMP-dependent signalling cascade acts mainly through cAMP dependent protein kinases (PKA) that are tetramers of two regulatory (that is, they bind cAMP) and two catalytic subunits, which may phosphorylate target proteins upon binding of two molecules of cAMP to each regulatory subunit, resulting in the release of catalytic subunits [7]. Four regulatory isoforms have been characterized in mammalian cells. According to cell types and functional conditions, these regulatory isoforms are differently expressed. Inside cells, regulatory isoforms can be partly diffusible and partly bound, mainly through PKA anchoring proteins called AKAPs, to cytoskeleton or intracellular organelles [8]. Differential expression as well as the ratio of the regulatory subunits of PKA RI/RII are important [9], since a high amount of RI has been detected in certain types of aggressive tumors [10]. Specifically, in the case of merlin, cAMP-dependent phosphorylation can greatly modify the behaviour of this oncosuppressor protein [11].

We have previously described the distribution of detergent-insoluble regulatory subunits of PKA in normal brain and found that it changes during brain development and in different disease models. The four regulatory subunits (RIAlpha, RIBeta, RIIAlpha, and RIIBeta) are expressed in different brain areas and even if present in the same cell, they are apparently located at different intracellular sites [12,13,14]. Interestingly, in different human brain tumors, we found different patterns of PKA distribution [15,16,17], supporting the idea that dysregulation of the cAMP pathway may serve as a diagnostic tool as well as a potentially druggable target [18,19]. 

The distribution of the regulatory and catalytic subunits has been examined in meningioma histological sections by immunofluorescence and fluorescent cAMP equilibrium binding, to unravel differences in comparison to the previously examined tumors. We also explored a gene expression database to highlight the correlation between PKA subunit or the expression of other genes involved in meningioma. The data presented here show a different pattern of distribution of PKA regulatory and catalytic subunits, compared to previously observed tumors. These data suggest a possible difference in biochemical activity due to different, never observed, cAMP-binding properties, compared to both normal brain and other brain tumors.

## 2. Results

### 2.1. Meningioma Tissue Binds cAMP

Meningioma specimens were obtained at surgery (see Appendix A) and were examined via immunofluorescence to reveal the presence of regulatory and catalytic subunits of PKA, followed by equilibrium binding of fluorescently-tagged 8-derivatives of cAMP. Representative images are shown in the figures and Appendix A. The conditions for optimal immunofluorescence were set on frozen samples: a mild fixation followed by permeabilization gave optimal results for PKA catalytic immunofluorescence (see Appendix A). However, all samples were examined with the different fixation protocols, and blind as concerns specific diagnosis, because diagnosis was not available at the time of examination. Under the same conditions, it was possible to observe labelling with fluorescent-tagged cAMP, which was due to specific cAMP binding, since it was completely abolished by the competitor non-fluorescent molecule 8Br-cAMP (Appendix A). The fluorescent cAMP labelling is not due to fixation artefacts, since it was present also in the unfixed tumor; see Appendix A, which shows another specimen, labelled with cAMP coupled to different fluorophores: similar results were obtained with fluorescein-, Alexa 488-, Alexa 555-, and rhodamine-tagged cAMP (see Materials and Methods), with diverse brightness due to different quantum yield of the fluorophores. The same structures were not labelled by another cyclic nucleotide, the fluorescein-tagged cGMP. In any case, the binding appears to be facilitated by washing or permeabilization, suggesting that it involves binding to insoluble structures.

### 2.2. PKA Regulatory and Catalytic Subunits Immunofluorescence 

In all meningioma samples, PKA RII immunofluorescence revealed strong punctuated labelling, apparently located outside the cell nuclei, as shown by DAPI nuclear counterstaining (Figure 1 and Figure 2 A–C).

After PKA RII immunofluorescence, incubation with fluorescent cAMP showed a nearly complete coincidence of the two signals (Figure 2 D–F): this condition was not observed previously in the normal brain or in other types of tumors. The lack of colocalization of fluorescent cAMP with PKA RI was surprising, since fluorescent cAMP-RI colocalized in the healthy brain tissue and also in other tumors, while in meningioma fluorescent cAMP mostly colocalized with PKA catalytic and RII subunit (see Figure 3 and Appendix A, which shows the original single-channel figures). 

### 2.3. PKA RI is Extremely Labile in Meningioma 

Another peculiarity of RI labelling in meningioma is its extreme lability, being present only in fixed but not permeabilized tissue. For both brain and other tumors processed in the same experimental session (for example, case 19 in reference [17], which was run with meningioma case 3), PKA RI immunolabelling was found to be present and colocalized with fluorescent cAMP, but not with PKA RII; moreover, both fluorescent cAMP and immunolabelling could be observed in permeabilized tissue. On the contrary, in the present series of meningioma samples, PKA RI immunolabelling was sensitive to permeabilization, disappearing in conditions in which fixation is followed by permeabilization, suggesting that PKA RI was present as a soluble pool. This inference is supported by preliminary observations of protein expression and partitioning in two cases, in which RI was detected in the soluble pool, while RII was present in the insoluble fraction, and CAT was present in both soluble and insoluble fractions (see Appendix A). To note, permeabilization does not affect fluorescent cAMP labelling, which appears to co-localize with both PKA RII and catalytic subunits, a condition not observed in healthy tissue or other tumors. A summary of the results obtained from immunofluorescence and equilibrium binding is presented in Table 1. 

### 2.4. Correlation of PKA Catalytic Subunit Expression Levels with Different Gene Products

Next, a database reporting the expression level of different genes in 68 meningioma cases was interrogated with BioGPS (see Material and Methods for details) to explore the relationship between PRKACA gene, coding for PKA CAT, and PKA regulatory subunits (PRKAR1A, PRKAR1B, PRKAR2A, and PRKAR2B) gene products. The expression level of PRKACA is positively, yet not significantly, correlated with PRKAR1B, PRKAR2A, and PRKAR2B (Pearson’s R = 0.1199, 0.1937 and 0.1659, respectively), while a significant negative correlation was detected between PRKAR1A and PRKACA (Pearson’s R = -0.2547, *p* < 0.05); see Figure 4.

Meningioma patients often present a loss of chromosome 22q, with inactivating neurofibromin2 (NF2, the gene coding for merlin) mutations in monosomic 22 specimens: NF2 expression level is positively and significantly correlated with PRKACA expression (R = 0.2994, *p* < 0.05), suggesting that both genes are downregulated in meningioma. Conversely, NF2 expression does not correlate significantly with any of the four PKA regulatory subunits expression levels (Table 2).

Subsequently, we focused on the four genes that are differentially over-expressed in recurrent meningioma: cell division cycle protein 2 homolog (CDC2, which codes for Cyclin-dependent kinase 1, CDK1), MLF1IP coding for Centromere Protein U (CENPU), Cyclin-dependent kinases regulatory subunit 2 (CKS2), and Protein Regulator of cytokinesis 1 (PRC1). The expression of all of these genes is significantly correlated in an inverse manner with PRKACA and PRKAR2B. There is only a slight yet significant positive correlation between MLF1IP and PRKAR2A expression (Table 2).

Lastly, we clustered the patients according to their status (dead N = 10 vs. alive N = 57, N = 1 patient status not determined), assuming that the dead patients had more aggressive tumors. However, no significant difference was detected between alive and dead patients in the expression levels for either PKA catalytic or regulatory subunits (Figure 5A), nor for the five recurrent meningioma-related proteins (Figure 5B).

## 3. Discussion

The cAMP signalling system acts mainly through PKA, which are present in all cells to mediate multiple information pathways, mainly through the expression of specific regulatory and catalytic subunits and through docking at precise subcellular locations. Regulatory subunits of PKA form homodimers, contributing to the specificity of the signalling system. We described the localization of the docked pool of the regulatory subunits of PKA in the brain of tetrapods, along with the changes that take place during ontogenetic development in the brain of mammals, birds, and reptiles [12,13,14,20,21,22], showing that homolog brain areas present a similar PKA distribution, related to cell type and developmental stage. This distribution may change in animal models of disease, like Parkinson’s disease and depression [23,24]. By using fluorescently-tagged cAMP, we observed that, while in vitro these molecules could bind to each form of soluble PKA regulatory subunit [25], in brain histological sections they could bind only to PKA RIAlpha [26], suggesting that the supramolecular organization of PKA docking complexes hindered the access of fluorescent molecules to PKA RII but not to RIAlpha. This also holds true for the brain of different species. Different issues were explored for tumors. In rodent and human glioma cells, we described a cluster of PKA RII localized in Golgi apparatus that could be targeted, resulting in apoptosis of tumor cells [15]. These data were further confirmed in a series of glioblastoma specimens in comparison to other 17 specimens of different tumors and grade [17], in which PKA RII was the most prominent labelling, with scarce, if any, PKA RI. In glioblastoma, fluorescent cAMP labelling was almost undetectable, with only scarce and sporadic labelling. A series of medulloblastoma was also examined: in this tumor, we observed the first deviation from PKA RI/fluorescent cAMP coincidence, normally found in the brain. In medulloblastoma, fluorescent cAMP labelled numerous structures that partly coincided with PKA catalytic subunit, but not with either PKA RI and RII, offering a new scenario for cAMP signalling within this tumor [16]. In the present work, we hypothesized that the PKA signalling pathways could be affected, since data from the literature highlight the cAMP system as a possible crossroad of many intracellular pathways.

Meningiomas have received much less attention than other CNS tumors due to their relative benign nature, despite being the most frequent intracranial tumors [4]; however, nearly 20% of meningiomas recur, without any clear molecular fingerprint for transformation, besides the common chromosome 22q loss, leading to NF2 inactivation [27]. At present, the WHO classification reports several meningioma subtypes, mainly pertaining to Grade I and three each to Grade II and III, based on histological appearance, without indicating molecular markers. A recent classification was proposed by a multicentric team based on genome-wide DNA methylation pattern for patient stratification [28]. Roughly 20% of meningiomas do not present obvious genetic alteration, while the most common genetic alteration is the inactivation of the NF2 gene coding for the protein Merlin. Usually, meningiomas present activation of receptors for multiple growth factors and their downstream signalling [29], with PDGF-beta overexpression driving malignant transformation in mice meningiomas [30]. However, several other pathways may be involved; for example, dysregulation of the MAPK signalling pathway [31,32], in particular, the p38 MAPK [33], or S6 kinase [34], or recently found mutations in in TRAF7, AKT1, KLF4, SMO, and PIK3CA [35], suggesting the use of targeted kinase inhibitors and immunotherapies [36]. At present, maximal surgical resection and radiation are the gold standard therapies, with alpha-interferon, VEGF inhibitors, and somatostatin receptor agonists used in recurrent meningiomas [37].

The cyclic AMP pathway is involved in the development of different brain tumors. Progression in some tumors may be linked to hyperactivation of the GalphaS pathway, which activates the PKA signalling, which in turn negatively regulates the Hippo pathway through blockade of the NF2-LAT-Yap signalling [38]. The involvement of the WNT signalling pathway has been recently recognized in meningiomas [39], with the WNT-coupled transcription factor FOXM1 being a key marker for aggressive meningiomas [40]; to note, the WNT pathway communicates with the G protein coupled receptor/PKA pathway to modulate beta-catenin phosphorylation and beta-catenin-dependent transcription [41]. 

As in other brain tumors, meningiomas also presented a higher level of the cAMP-degrading enzyme phosphodiesterase 4A [42], suggesting that targeting this pathway could be a feasible option for the clinical management of recurrent meningiomas. This idea is supported by microarray expression data, showing differential expression of phosphodiesterase 1C in different grade meningiomas [43]. Phosphodiesterases are also involved in tumorigenesis in Neurofibromatosis type 1 (NF1) models [44], and phosphodiesterase 5 blockade has resulted in breast cancer stem cell elimination through the activation of PKA [45]. By analysing the kinome and phosphoproteomic profiles of meningioma of different grades, it was recently shown that the A-kinase anchoring protein 12, a docking protein for PKA RII, was downregulated according to increasing malignancy grade, by comparing Grades II and III to Grade I meningiomas; this highlights AKAP12 as a prognostic marker for invasiveness and a central regulator in meningioma [46]. Another meningioma series reports significant upregulation of AKAP12 and phospho-AKAP12 in Grade I and II meningiomas compared to meningeal tissue, while a non-significant increase is reported for Grade III meningiomas [47]; in any case, the involvement of AKAP12 seems warranted. The relationships between the regulatory type RI and RII with AKAP12 will be explored in our future works. Meningioma cells were demonstrated to react to cAMP or cAMP-elevating agents with a decrease in cell growth rate, an effect partly mediated via the secretion of interleukin 6, which acts as an inhibitory factor [48]. On the contrary, increase of cell growth may be induced by somatostatin, since meningiomas mostly express somatostatin receptors that repress the formation of cAMP [49]. Interestingly, amplification of the long arm of chromosome 17 was found in anaplastic meningiomas, in proximity of the locus hosting PRKAR1A gene, encoding for PKA RIAlpha, suggesting a possible link to its expression [50].

Prompted by the differences in the localization of PKA catalytic and regulatory subunits, we explored the gene expression profile of a series of meningioma patients, to reveal possible relationships between the PKA isoforms and other genes specifically affected in meningioma. At variance with the normal brain and the malignant brain tumor glioblastoma (see Figure S1A in [17] for a comparison), in meningioma, PRKAR1A appears the most expressed PKA regulatory isoform. This is also supported by a proteomic study comparing different grade meningiomas to normal meninges [47], which reports a significant increase in non-phosphorylated RI for Grade II and III meningiomas and in phosphorylated RI for all grades, together with increases in phosphorylated proteins RIIA, RIIB, and CAT in Grade III. Interestingly, recurrent mutation in PRKAR1A has been reported in few meningioma cases with non-mutated NF2 gene [51]. As shown in the present data, the distribution of RI in meningiomas is different from what can be appreciated in the brain or in other tumors like glioblastoma or medulloblastoma [16,17]. Noteworthy, PRKAR1A is the only regulatory subunit gene whose expression correlates significantly, albeit negatively, with PKA catalytic subunits. This observation suggests that PRKACA expression may be negatively regulated in meningioma. The positive correlation of PRKACA with NF2 gene expression [27], which is downregulated in meningioma, may support this view, as well as the negative correlation of PRKACA expression with the four upregulated genes typically found in recurrent meningiomas (CDC2, MLF1IP, CKS2, and PRC1) [27]. These data prompt further analysis and validation with protein quantification, because of a possible non-linearity in protein translation or subsequent modifications in the protein itself or in protein turnover.

Our present data add another piece to the biomolecular characterization of meningioma, further supporting the centrality of the cAMP signalling pathway in the development of these tumors. Here, we show that the meningioma tissue presents organization of PKA that has different features from what can be observed in healthy brain or different tumors of the brain of different origin, like medulloblastoma and glioblastoma. In particular, PKA RI appears mostly present in soluble form, besides being the most expressed PKA regulatory subunit. The coincidence of fluorescent cAMP labelling with PKA RII and catalytic subunits suggest a different supramolecular organization that could be exploitable as a possible therapeutic target. Moreover, the use of fluorescent cAMP labelling could be of interest for diagnostic purposes during surgery, given that it also works in few minutes in unfixed tissue. Hence, the present data highlight three molecular properties (e.g. RI lability, CAT/RII colocalization, and RII binding cAMP 8-derivatives in situ) that could not be predicted or demonstrated by other approaches.

## 4. Materials and Methods 

### 4.1. Patients

This study was performed according to the 1964 Declaration of Helsinki and its later amendments. It conformed to Italian legislation and was approved by the institutional Ethical Committee (*Comitato Etico per la Sperimentazione — Azienda Ospedaliera di Padova* protocol number 1883P). Patients (see Appendix A for details) gave their written informed consent and were recruited at the University of Padova Medical School. During surgery for tumor removal, one fragment of the tumor tissue was excised and immediately frozen by dipping in liquid nitrogen. Tissues were stored at −70 °C until sectioning.

### 4.2. Tissue Processing

The chemicals were from Sigma (Milan, Italy), unless otherwise stated. Immunofluorescence and equilibrium binding protocols conform to previously published methods [16,17]. Tissues were cut on a cryostat at 20 μm. Serial sections were air dried and fixed with four different protocols: (1) Mild fixation, 1 minute in formalin 5% in phosphate-buffered saline (PBS) at 37°C; (2) mild fixation followed by permeabilization, 1 minute in formalin 5% at 37°C then 30’ in Triton X-100 2% in PBS; (3) formalin 5% 1 hour at room temperature, then Triton X-100 2% 30’; and (4) Triton X-100 2% 30’, then formalin 5% 1’ at 37°C. The primary antibodies were incubated overnight. The following antibodies were used [15]: Anti PKA RIIA (Santa Cruz Biotechnology, cross-reactive for both RII alpha and beta isoforms) 1:200; Anti PKA RIB (Santa Cruz Biotechnology, cross-reactive for both RI alpha and beta isoforms) 1:200; and Anti PKA catalytic subunit 1:200 (Santa Cruz Biotechnology, cross-reactive for all isoforms). Secondary antibody (1:200 on tissue) was incubated for 30 min at 37°C: anti-rabbit IgG Alexafluor 594-conjugate (Molecular Probes-Invitrogen, Milan, Italy), or anti-rabbit IgG fluorescein-conjugate (Sigma, Milan, Italy). Cell nuclei were counterstained with DAPI (Sigma, Milan, Italy). Positive and negative controls were included in each staining session: positive controls were mouse brain sections, whose labeling pattern is known [12,13,14]; negative controls were processed omitting the primary antibody, or incubated with normal rabbit serum.

For the colocalization experiments, some sections were incubated for 10 minutes at room temperature in PBS containing one of the following fluorescent nucleotides: 300 nM 8-(5-thioacetamidofluorescein)-adenosine 3’,5’-cyclic monophosphate (SAF-cAMP), or with 300 nM 8-(5-thioacetamidofluorescein)-guanosin 3’,5’-cyclic monophosphate (SAF-cGMP) or with 8-(5-thioacetamidotetramethylrhodamine)-adenosine 3’,5’-cyclic monophosphate (SAR-cAMP), or with 250 nM 8-(2-fluoresceinylthioureidoaminoethylthio)-adenosine 3’,5’ -cyclic monophosphate (8-Fluo-cAMP), or with 100 nM 8-(Alexa488)- adenosine 3’,5’-cyclic monophosphate (Alexa488-cAMP) or 8-(Alexa555)- adenosine 3’,5’-cyclic monophosphate (Alexa555-cAMP), as previously described [26]. SAF-cAMP, SAF-cGMP, and SAR-cAMP were synthesized [52], Alexa488-cAMP and Alexa 555-cAMP were from Molecular Probes (Eugene, OR), 8-Fluo-cAMP was obtained from BioLog (Germany). They are all readily displaced by 50 μM 8-Br-cAMP, resulting in specific abolition of the fluorescent cAMP labelling (see Appendix A). After immunofluorescence, the sections were counterstained with hematoxylin–eosin. 

### 4.3. Image Analysis

Slides were observed with a Leica epifluorescence microscope (20×, 40×, and 100× objectives). Immunofluorescence and equilibrium binding labelling was independently graded during observation at the microscope by two observers on a semiquantitative scale. Images were acquired with the resident software at 768 × 582 pixels with an RGB color digital camera, using the same parameters within each experiment. The images were used unaltered, unless otherwise stated. They were superimposed using Graphic Converter 9 and set up in Powerpoint 16.16.6. For determining colocalization, images were analyzed with ImageJ software. Each image was split on the three channels and the maxima (pixels intensity at least 10 AU above adjacent background) were marked and counted, then red and green images were superimposed, and the maxima were evaluated as being superimposed (if they were coincidental) or not, in order to obtain the number of green only, red only, and coincidental maxima, from which the percentage was calculated.

### 4.4. Expression Data Analysis

Data obtained with Affymetrix U133 Plus 2.0 expression arrays from 68 meningioma patients (age range 32-89 years, both genders), initially reported in [27] were interrogated with BioGPS (www.biogps.org) [53,54,55]. Histograms were generated and expression levels was correlated with Pearson’s R statistic, while expression levels between dead and alive patients were compared with T-test using R software. Significant level was set at *p* < 0.05.

## 5. Conclusions

Meningioma has no clear molecular landmarks. Despite presenting few mutations, some data indicate that the cAMP pathway may be involved in dysregulated growth of meningioma cells. Here, we found that PKA regulatory subunits are distributed differently in meningioma, compared to a healthy brain and other tumors, and present a different cAMP-binding activity, which may be of interest for therapeutic or diagnostic purposes.

## Figures and Tables

**Figure 1 cancers-11-01686-f001:**
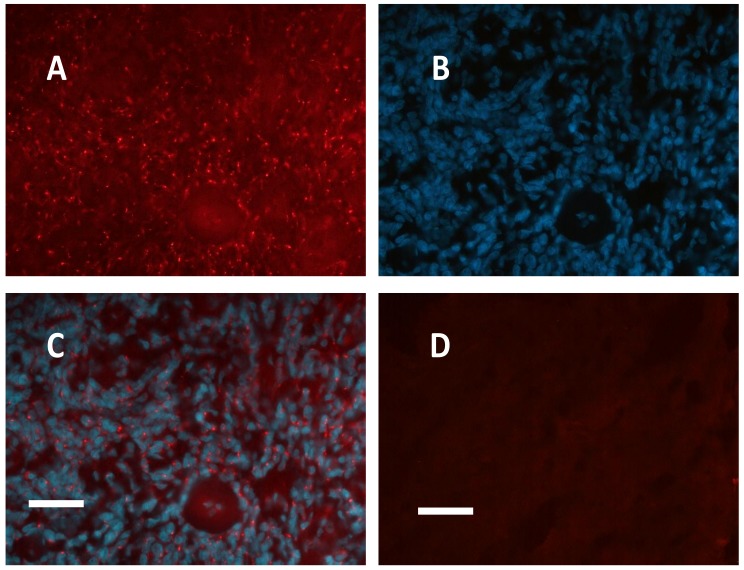
Case 2, meningioma with sclerosis areas. (**A**): PKA RII immunofluorescence. (**B**) Same photographic field, nuclei are labelled with DAPI. (**C**) Merge of (**A**) and (**B**). (**D**) Negative control, omission of primary antibody in an adjacent section (incubation with only secondary antibody, red-labelled) results in no red signals. Unaltered images. **A** and **D** taken with the same filters setting and same exposure time from adjacent sections. Objective: 40x. Bar = 25 μm.

**Figure 2 cancers-11-01686-f002:**
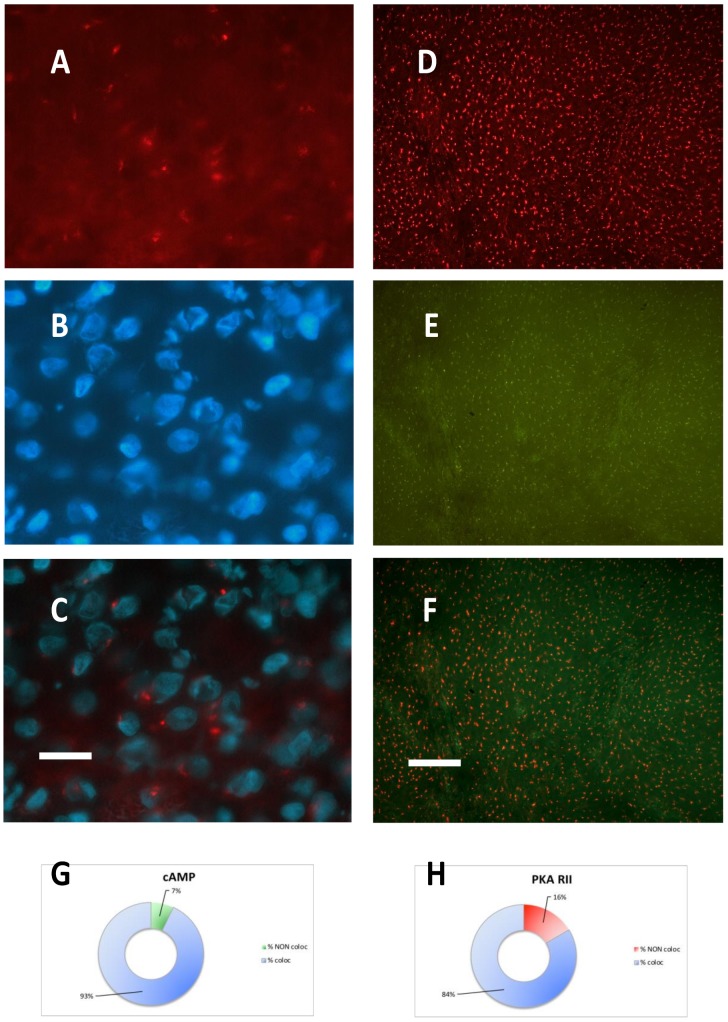
Case 3 representative images. (**A**) PKA RII immunofluorescence. (**B**) Same field, DAPI nuclear counterstaining. (**C**) Merging of (**A**) and (**B**). (**D**) Low power image of PKA RII immunofluorescence. (**E**) Same field, SAF-cAMP labelling. (**F**) Merging of (**D**) and (**E**); the orange color indicates coincidence of the red and green signals. (**G**) Percentage of colocalization (blue) or non-colocalization (green) of cAMP with PKA RII, relative to Figure 2 (**F**). (**H**) Percentage of colocalization (blue) or non-colocalization (red) of PKA RII with cAMP, relative to Figure 2 (**F**). (**A**–**C**): 100x objective, bar = 10 μm. (**D**–**F**): 20x objective, bar = 50 μm. Unaltered images.

**Figure 3 cancers-11-01686-f003:**
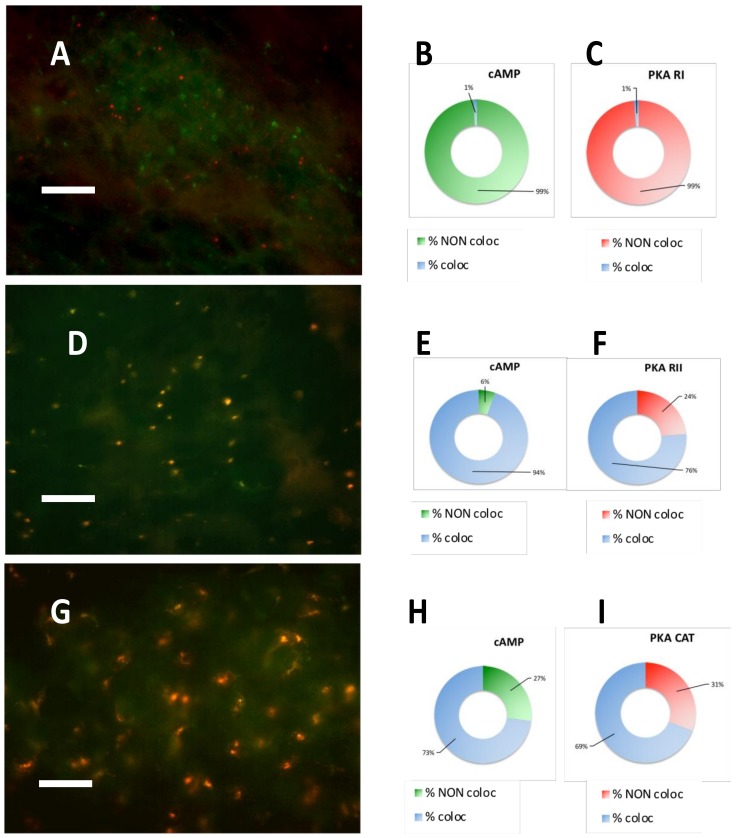
Case 5 representative images. (**A**) Merging of PKA RI immunofluorescence (red) and SAF-cAMP labelling (green): the signals are clearly separated (see original single channel images in Appendix A). (**B**) Percentage of colocalization (blue) or non-colocalization (green) of cAMP with PKA RI, relative to Figure 3A. (**C**) Percentage of colocalization (blue) or non-colocalization (red) of PKA RI with cAMP, relative to Figure 3A. (**D**) Merging of PKA RII immunofluorescence (red) and SAF-cAMP labelling (green): the signals coincide as shown by the yellowish labelling (see original single channel images in Appendix A). (**E**) Percentage of colocalization (blue) or non-colocalization (green) of cAMP with PKA RII, relative to Figure 3D. (**F**) Percentage of colocalization (blue) or non-colocalization (red) of PKA RII with cAMP, relative to Figure 3D. (**G**) Merging of PKA catalytic subunit immunofluorescence (red) and SAF-cAMP labelling (green): the signals coincide, as shown by the orange-yellowish color (see original single channel images in Appendix A). (**H**) Percentage of colocalization (blue) or non-colocalization (green) of cAMP with PKA catalytic subunit (CAT), relative to Figure 3G. (**I**) Percentage of colocalization (blue) or non-colocalization (red) of PKA catalytic subunit with cAMP, relative to Figure 3G. 100x objective, bar = 10 μm. Unaltered images.

**Figure 4 cancers-11-01686-f004:**
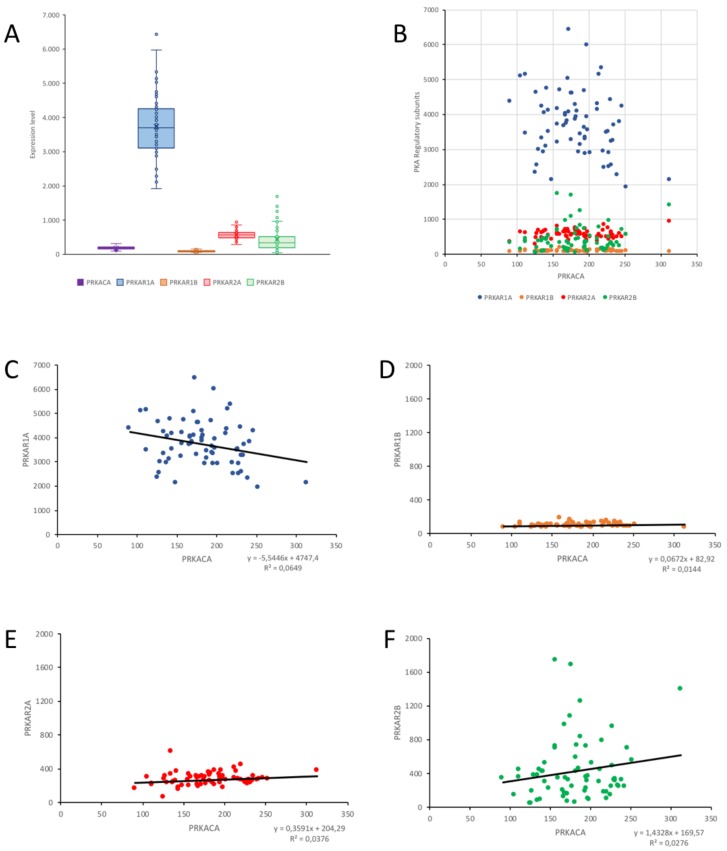
Gene expression of PKA catalytic and regulatory subunits in a cohort of 68 meningioma patients. (**A**) Gene expression level for PRKACA, PRKAR1A, PRKAR1B, PRKAR2A, and PRKAR2B. (**B**) Comparative view of the correlation between PRKACA expression and the four genes coding for the PKA regulatory subunits. (**C**–**F**) Correlation of the expression of PRKACA with each of the single genes for PKA regulatory subunits shown in (**B**). (**C**) A significant negative correlation is present between PRKACA and PRKAR1A expression, while weak, non-significant correlations are present between PRKACA and PRKAR1B (**D**), PRKAR2A (**E**), and PRKAR2B (**F**).

**Figure 5 cancers-11-01686-f005:**
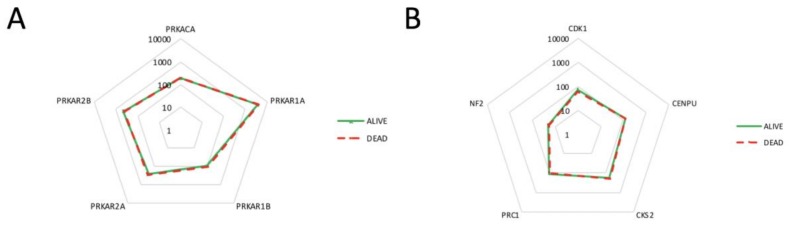
Expression levels of different genes in patients still alive or dead at the end of the study. (**A**) The expression level of the PKA regulatory and catalytic genes does not differ between dead and alive patients. (**B**) The expression level of five meningioma-related genes does not differ between dead and alive patients. Log10 axes.

**Table 1 cancers-11-01686-t001:** Summary of results obtained with the fixation protocol that gives optimal results for each antigen. The intensity of equilibrium binding was evaluated in the best labelled immunofluorescent sections. Grading is a mean of values obtained in different repetitions of the experiment by two observers that independently scored the slides. IF: immunofluorescence. NA: not available, slices lost during processing. Grading: −: negative. −/+: some positive trials, some negative. +: faint labelling. ++: moderate. +++: intense labelling. The relative intensity was evaluated for each case.

CASE N°	cAMP equilibrium binding	PKA RI IF	PKA RII IF	PKA CATALYTIC IF
1	+	+ Mostly on vessels	+++	+++
2	+/− Only in some zones	+	+++ Only in some zones	+++ Only in some zones
3	+++	+	+++	+++
4	++	+	+++	+++
5	++	++	++	++
6	+	+	++	++
7	+	NA	NA	++
8	++	+	+++	+++
9	-	+	++ Only in some zones	++ Only in some zones
10	+	+ Only in some zones	+++	+++
11	+/−	-	+++	+++
12	+++	+	+++	+++
13	+++	-	+++	+++

**Table 2 cancers-11-01686-t002:** Correlation of PRKA subunits’ gene expression levels with meningioma-related genes (Pearson’s R values. Significant correlations are in bold and marked by * *p* < 0.05, ** *p* < 0.02, *** *p* < 0.005).

	PRKACA	PRKAR1A	PRKAR1B	PRKAR2A	PRKAR2B
**NF2**	**0.2994 ****	0.0679	−0.0406	0.0965	0.1524
**CDC2**	**−0.2973 ****	0.0518	−0.0246	0.1128	**−0.3763 *****
**MLF1IP**	**−0.2908 ****	0.0363	−0.0844	**0.2692 ***	**−0.3098 ****
**CKS2**	**−0.3188 ****	−0.1402	−0.1074	0.0945	**−0.3754 *****
**PRC1**	**−0.3622 *****	−0.0497	−0.0640	0.1630	**−0.3563 *****

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
