# Peer review of "Protein Kinase A Distribution in Meningioma"

_cancers, 2019, doi:10.3390/cancers11111686_

Round 1
Reviewer 1 Report
The authors have responded in satisfactory manner to prior recommendations.
Author Response
Thank you for the positive comment.
Reviewer 2 Report
This manuscript has been improved.
Author Response
Thank you for the positive comment.
Reviewer 3 Report
The authors have made some improvements to this version of the manuscript and they added more data concerning the correlation of the expression levels of the different isoforms of PKA catalytic subunits and regulatory subunits in a cohort of 68 meningioma patients. However, there are still a number of issues that need to be addressed before it is acceptable for publication. Some of the issues were pointed out in my report to the previous version of the manuscript. These issues are:
The authors need to perform Western blot analysis of control tissues and meningioma tissue biopsies or meningioma cell lines to examine the expression levels and integrity of PKA R1 and RII. The lack of overlap of the cAMP staining pattern and the RI immunostain pattern in meningioma can be caused by the very low level of RI protein expression and proteolytic processing of the RI to form truncated fragment with a much lower affinity for cAMP. Results of Western blot analysis will clarify this issue. I would like to point out that the authors did Western blot analysis of R1 and RII in a paper on glioblastoma and brain tissues published in Cancer in 2017. The authors provided additional data demonstrating a significant negative correlation of PRKAR1A and PRKACA mRNA levels in 68 meningioma cases. Against, western blot analysis can validate this finding. If the authors have trouble obtaining enough primary cancer tissues to perform western blot analysis, they should at least perform the experiment with meningioma cell lines. In the supplementary information, the authors gave a list of 13 meningioma cases. They should generate a table and indicate the staining patterns and/or expression levels of cAMP and all isoforms of PKA C and R subunits. The authors need to cite and discuss the findings of the manuscript published by Dunn, et al in EBiomedicine (Dunn, J., et al. (2019) “Protoemic analysis discovers…” EBiomedcine 40:77-91). Parada, et al. reported downregulation of AKAP12 as a regulator of aggressiveness of meningioma. How does AKAP12 downregulation relates to the paranuclear localisation of the RII subunit presented by the authors in Figures 1 and 2 of the manuscript. I raised a similar question in my report to the previous version of the manuscript. The manuscript still requires further editing. For example, there is only one sentence in a paragraph (lines 60-61) of the Introduction section.
Round 2
Reviewer 3 Report
The authors have addressed all my concerns. The manuscript requires minor revision to supplementary figure S6 before it is acceptable for publication (see below).
The authors need to add moleuclar mass labels to the western blots in Figure S6.
Author Response
Dear Editor,
we apologize for our inaccuracy and thank the Reviewer for having suggested the improvement for Supplementary Figure 6. The figure has been amended as requested by adding the apparent molecular mass of standards.
Thank you for your comprehension,
Best regards,
Carla Mucignat-Caretta
